# Oncogenic Virome Benefits from the Different Vaginal Microbiome-Immune Axes

**DOI:** 10.3390/microorganisms7100414

**Published:** 2019-10-01

**Authors:** Giuseppina Campisciano, Tarik Gheit, Francesco De Seta, Carolina Cason, Nunzia Zanotta, Serena Delbue, Giuseppe Ricci, Pasquale Ferrante, Massimo Tommasino, Manola Comar

**Affiliations:** 1Advanced Laboratory of Translational Microbiology, Institute for maternal and child health “IRCCS Burlo Garofolo”, Via dell’Istria 65, 34137 Trieste, Italy; giusi.campisciano@burlo.trieste.it (G.C.); francesco.deseta@burlo.trieste.it (F.D.S.); carolina.cason@burlo.trieste.it (C.C.); nunzia.zanotta@burlo.trieste.it (N.Z.); giuseppe.ricci@burlo.trieste.it (G.R.); 2Infections and Cancer Biology Group, IARC, 150 Cours Albert Thomas, 69008 Lyon, France; gheitt@iarc.fr (T.G.); tommasinom@iarc.fr (M.T.); 3Obstetrics and Gynecology, Institute for maternal and child health “IRCCS Burlo Garofolo”, Via dell’Istria 65, 34137 Trieste, Italy; 4Department of Medical Sciences, UNITS Cattinara Hospital, Strada di Fiume 447, 34149 Trieste, Italy; 5Laboratory of Translational Research, Department of Biomedical, Surgical and Dental Sciences, University of Milano, Via Carlo Pascal, 36, 20133 Milano, Italy; serena.delbue@unimi.it (S.D.); pasquale.ferrante@unimi.it (P.F.)

**Keywords:** virome, bacteriome, immune response, vagitypes

## Abstract

The picture of dynamic interaction between oncogenic viruses and the vaginal bacteria-immune host milieu is incomplete. We evaluated the impact of *Polyomaviridae*, *Papillomaviridae*, and *Herpesviridae* oncoviruses on the vaginal Community State Types (CSTs) and host immune response in reproductive-age women. In our cohort, only *Polyomaviridae* and *Papillomaviridae* were detected and were associated with changes in the resident bacteria of CST I and IV (*p* < 0.05). *Lactobacillus*
*crispatus* increased in CST I while *Prevotella timonensis* and *Sneathia sanguinegens* increased in CST IV. Conversely, CST II and III showed an alteration of the immune response, with the decrease of Eotaxin, MCP-1, IL-7, IL-9, and IL-15 (*p* < 0.05), leading to reduced antiviral efficacy. An efficient viral clearance was observed only in women from CST I, dominated by *Lactobacillus crispatus*. Our in vivo study begins to address the knowledge gap with respect to the role of vaginal bacteria and immune response in susceptibility to oncoviral infections.

## 1. Introduction

The mutualism between the vaginal mucosa and its bacterial microbiome is a key point of the physiologic condition of the female reproductive tract [1].

Cumulative pieces of evidence have shown that, alongside bacteria, viruses are a constitutive part of the vaginal microbiome and that the virome is composed by both eukaryotic and prokaryotic viruses. As the virome study methodologies improve, eukaryotic viruses causing latent or asymptomatic infections are being recognized as the more prominent members of the vaginal microbiome, suggesting an intimate relationship with both resident bacteria and host susceptibility and immune response to viral infections [2].

Though to date, metagenomic studies on this complex interaction are far from reaching a complete understanding, the most intriguing aspect is whether and how bacterial community may promote or inhibit viral pathogenesis [3]. A portion ranging from 15%–20% of all human cancers is caused by oncogenic viruses [4]. The oncogenic viruses infecting the cervical tract can cause asymptomatic latent infections and induce cellular transformation after many decades. Although each virus has its own specific mechanism for promoting carcinogenesis, the ability of these viruses to establish a latent persistent infection is critical to incidental viral tumorigenesis [5].

The vaginal dysbiosis of the mucosa-associated microbiota seems to represent a risk factor acting as an enhancer of the tumorigenic potential through mechanisms including mucosal barrier failure and inflammation [3]. These events increase the rate of a successful viral infection, enabling viral survival in the cells through an asymptomatic and persistent nonlytic infection which determines a very subtle inflammatory response and positively influences the induced tumorigenesis in immunosuppressed women [2,6].

Since a persistent infection enables virus-induced tumorigenesis, we can develop new approaches for preventing and treating malignancies by characterizing the host environment by which tumor viruses achieve infection.

Wojciech Kwasniewski et al. hypothesized that a disturbed heterogeneity of the vaginal microbiome is associated with Human Papilloma Virus (HPV)-induced carcinogenesis, identifying the vaginal dysbiosis as a putative and potentially modifiable factor for HPV [7]. Moreover, a very recent meta-analysis supported the hypothesis of a causal link between vaginal dysbiosis and cervical cancer at several disease stages, affirming that a *Lactobacillus* dominating the vaginal microbiota might reduce HPV-related disease burden or increase the viral clearance [8].

The “omics” technologies allowed us to discover that several divergent oncogenic viruses promote tumorigenesis through shared host cell targets and pathways modulating the surrounding microenvironment [9]. During the last few years, several observational studies reported oncogenic viruses infecting the vaginal mucosa as a possible trigger for HPV in women of reproductive age, including viral species belonging to the *Polyomaviridae* and *Herpesviridae* families. Despite this possible finding, current knowledge on dynamic interaction with vaginal microbiome and host immune response is scarce [10,11].

In this study, we integrated metagenomics and immune quantitative approaches to provide, to our knowledge, the first in vivo view of the interaction of *Papillomavirus*, *Polyomavirus*, and *Herpesvirus* oncogenic viruses with the vaginal community state types (CSTs) and local immune environments before any clinical evidence of the cellular transformation is obtained.

## 2. Materials and Methods

### 2.1. Demographics of the Studied Cohort

During the first two months of 2018, 90 vaginal swabs from Caucasian immunocompetent women of reproductive age (20–40) who visited the gynecological ambulatory of the IRCCS Burlo Garofolo, Trieste, Italy, for urogenital complaints, were retrospectively selected on the basis of the following exclusion criteria: pregnancy, menopause, 5–7 days after menstruation, antibiotic/probiotic therapy, oral contraceptive use or hormonal replacement therapy, smoking, endocrine diseases, history of cancer, current laboratory detection of STIs pathogens, HIV, and cervical-vaginal lesions.

After 8 months from the laboratory diagnosis, following the indications of the clinical protocol, an additional vaginal swab was collected only from women who tested positive for viral infection (*n* = 29).

Vaginal specimens were collected from each woman following a standard procedure using a polyethylene Cervex brush device (Rovers Medical Devices B.V., Oss, The Netherlands). The swab was suspended in 1.5 mL of TE buffer for microbiome assessment and immune factors dosage, and stored at −80 °C. 

### 2.2. DNA Extraction and NGS Library Preparation

DNA was extracted from the swabs using the NucliSENS^®^ easyMAG^®^ system (BioMèrieux, Gorman, North Carolina, CA, USA), following the manufacturer’s instructions, starting from 500 µL of sample and with an elution volume of 50 µL. All DNA samples were stored at −80 °C prior to further processing.

Firstly, a 500 base pair region of the V1-V3 portion of the 16S rRNA gene was amplified and subsequently the 200 base pair region of the V3 portion was amplified as well, as described elsewhere [12]. The V3 amplicon was used for template preparation by the Ion PGM Hi-Q View kit on the Ion OneTouch™ 2 System (Life Technologies, Gran Island, New York, NY, USA) and sequenced using the Ion PGM Hi-Q View sequencing kit (Life Technologies, New York, NY, USA) with the Ion PGM™ System technology. Negative controls, including a no template control, were processed with the clinical samples.

### 2.3. Big Data Processing

Quality filtering was performed using the software QIIME (Quantitative Insights Into Microbial Ecology), version 1.9.1 [13]. Sequences with ambiguous bases and/or with mean Phred quality scores ≤20 were discarded. Taxonomic identities were assigned using the Vaginal 16S rRNA gene Reference Database, which was constructed by Fettweis et al. [14], using open-reference OTU (Observational Taxonomic Unit) picking with a uclust clustering tool. To reduce the risk of including OTUs that were PCR artifacts, all OTUs that occurred in only one sample were removed. To control for differences in sequencing depth between samples, we normalized the read counts by rarefying the otu table biom to a depth of 5000 reads/sample.

### 2.4. Oncogenic Virome Analysis

The identification of infectious agents was performed using a highly sensitive species-specific multiplex genotyping assay, combined with multiplex polymerase chain reaction (PCR), and a bead-based Luminex technology (Luminex Corp., Austin, TX, USA). Multiplex type-specific PCR (QIAGEN Multiplex PCR kit, Venlo, Netherlands) used primers specific to detect the presence of 21 mucosal HPV types (HPV-6, -11, -16, -18, -26, -31, -33, -35, -39, -45, -51, -52, -53, -56, -58, -59, -66, -68, -70, -73, and -82), 13 *Polyomaviruses* (BKPyV, JCPyV, KIPyV, MCPyV, WUPyV, TSPyV, HPyV6, HPyV7, HPyV9, HPyV10, HPyV12, LIPyV, and SV40), and 10 *Herpesviruses* (CMV, EBV1, EBV2, HSV1, HSV2, HHV3, HHV6A, HHV6B, HHHV7, and HHV8) [15,16], respectively. Two primer sets for the amplification of β-globin were used as a positive control for the assessment of template DNA quality. Briefly, the PCR products were generated, denatured, and hybridized to the bead-coupled probes in 96-well plates, as previously described [17,18]. Subsequently, the beads were analyzed in the Luminex reader, the results were expressed as the median fluorescence intensity (MFI). The cut-off was computed by adding 5 MFI to 1.1× the median background value.

### 2.5. Dosage of the Immune Soluble Factors

The concentrations of 48 cytokines, chemokines, and growth factors were analyzed by a quantitative cytokine assay by means of the Bioplex Pro™ human cytokine standard 27-plex and 21-plex panels based on xMAP technology (Bio-Rad Laboratory, Hercules, CA, USA). Data were collected and analyzed using a Bio-Rad BioPlex 200 instrument equipped with Bio-Plex Manager software version 6.0 (Bio-Rad Laboratory, Hercules, CA, USA). The immunoassay data were expressed in terms of mg protein/mL estimated from Bradford protein concentration in respective samples using Bio-Rad microtiter microassay kit (Bio-Rad Laboratory, Hercules, CA, USA) and human serum albumin (Sigma Chemical, St. Louis, MO, USA), as recently published [12].

### 2.6. Statistical Analysis

Using QIIME 1.9.1, Chao1, Observed OTUs, and Phylogenetic diversity (PD) whole tree metrics were calculated to assess the alpha diversity and compared by means of a nonparametric T-test.

To highlight the differences in the microbial composition we applied a nonparametric T-test for the pairwise comparisons.

To test the differences in the immune soluble factors, GraphPad Prism (v. 5, San Diego, CA USA) was used. Specifically, the Kruskal-Wallis one-way analysis of variance was used for comparisons between groups. When a significant *p*-value was observed (*p* < 0.05), a multiple comparison test was used to determine which groups were different.

### 2.7. Data Availability

The dataset supporting the conclusions of this article has been uploaded to the National Center for Biotechnology Information (NCBI) Sequence Read Archive (SRA) under the project number SRP152778.

## 3. Results

### 3.1. The Vaginal Bacteriome According to the Community State Types

Bacterial microbiome composition was evaluated in 90 vaginal swabs through sequencing of the barcoded V3 region of the 16S rRNA gene, and the swabs were clustered into four community state types (CSTs) based on the predominant *Lactobacillus* species. Four samples provided an exception since the microbiome was characterized by a massive colonization of *Bifidobacteria*, classified as B group for two of them, and the remaining two, classified as Mixed CST, showed two dominant *Lactobacilli* at equal amounts (in one sample *L. crispatus* and *L. gasseri* and in two samples *L. crispatus* and *L. iners*).

We obtained a total of 5,683,658 reads (range 1278–190,380) and a total number of observed OTUs of 11,168. For the analyses, we rarefied the otu_table.biom to a depth of 5000 reads/sample, excluding four samples. The two negative controls did not produce an output after the quality filtering.

Figure 1 shows the identified microbiome, sub-grouped according to the presence (black cells) or absence (white cells) of oncogenic viral strains.

Figure 2 highlights the mean relative abundance of the dominant bacteria of each CST and the significantly varied bacteria among CSTs. In CST I, the mean relative abundance of *Lactobacillus crispatus* was 72%, while in CST II, the average relative abundance of *Lactobacillus gasseri* was 58%. In CST III, *Lactobacillus iners* was 94%, while in CST IV, *G. vaginalis* was 10%. L. *iners* (40%) and *B. breve* (58%) were the dominant species in Mixed and B groups, respectively.

CST IV, as expected, showed the highest bacterial heterogeneity, including the colonization by a high rate of Gram-positive microorganisms (26%) dominated by *Streptococcus* spp. (12%) and by Gram-negative species such as *Atopobium vaginae* (7%) and *Escherichia fergusonii* (5%), compatible with a status of dysbiosis and biofilm production. Conversely, the CST III was the most homogeneous group, showing only the 6% of bacterial species that were different from *L. iners* (94%), underscoring its ability to colonize the vaginal epithelium both in eubiotic and dysbiotic conditions. Indeed, the alpha diversity showed a significant decrease only for samples from CST III (*p* < 0.005), showing the most homogeneous bacterial composition. When viral infected CSTs were compared with the corresponding negative ones, no statistical variation was observed (Table 1).

*Ureaplasma parvum* has been detected only in the CST I and CST II where, in this last group, *U. parvum* showed the highest rate of colonization. Moreover, *Gardnerella vaginalis* has been detected in all samples, but with a relative amount inversely proportional to that of *Lactobacilli* spp. (Figure 2).

### 3.2. Papillomavirus, Polyomavirus and Herpesvirus Oncogenic Virome According to CSTs

The virome analysis highlighted that a small cluster of oncogenic viruses, including *Papillomaviridae* and *Polyomaviridae* family strains, were present in our series and they were not preferentially identified in a given CST (Figure 3). We did not identify *Herpesviridae* family strains in our cohort. In 29/86 (33.7%) samples, DNA of oncogenic viruses was identified. MCPyV was the *Polyomavirus* most frequently detected, while the HPV genotypes identified mostly belongs to the High-Risk HPV genotypes. The highest rate of infection of this series was measured in CST III (43%, 12/28) and in CST I (38%, 7/18) than in CST IV (27%, 6/22) and CST II (23%, 3/13). HPV DNA was more frequently associated with CST IV (5/6) followed by CST III (8/28) while regarding *Polyomaviruses*, the highest rate of infection was recovered in CST III (3/3) and CST I (5/7).

The overall distribution highlighted the strong presence of *Polyomaviruses*, specifically of JCPyV and MCPyV, as unique or co-infections in all CST groups. In detail, in CST I, 2/7 samples were infected with HPV, 1/7 with HPV/JCPyV, 1/7 with JCPyV/MCPyV and 3/7 with MCPyV. Among CST II samples, 1/3 showed HPV/MCPyV and 2/3 MCPyV. Within the CST III, 6/12 were infected with HPV, 1/12 with HPV/JCPyV, 1/12 with HPV/MCPyV, 2/12 with JCPyV and MCPyV. In CST IV, 3/6 samples were infected with HPV, 1/6 with HPV/JCPyV, 1/6 HPV/MCPyV and 1/6 with MCPyV. 

One of the two remaining samples showing a massive presence of *Bifidobacteria* indicated a positive sign positive for MCPyV. No viral infections were detected in the samples defined as mixed CST.

### 3.3. Oncogenic Viral Infections and CSTs: Bacterial and Immune Responses

The effect of the oncogenic viral infections on the vaginal microbial composition and/or immune environment perturbations was analyzed for each CST group and described in Figure 4.

Considering oncogenic viruses as a variable, we observed a significant modulation only for dominant bacteria characterizing CST I and IV samples (*p* < 0.05). Specifically, *L. crispatus* increased in infected samples from CST I, showing a high rate infection with *Polyomavirus*, while *Prevotella timonensis* and *Sneathia sanguinegens* increased in infected samples from CST IV, characterized by a high rate of HPV infection.

As detailed in Table 2, the basal status of the immune soluble environment for each CST group in absence or presence of viral infections has been characterized and compared. Among the tested 47 immune soluble factors, several cytokines involved in inflammatory and anti-tumoral activities showed a significant decrease in positive samples for oncogenic viruses. Among samples from CST II and CST III, a significantly decreased concentration of the cytokines IL-7, Eotaxin and MCP1 was detected in CST II while, in CST III, a decreased concentration was detected for IL-15 and IL-9 (*p* < 0.05). Conversely, infected samples from CST I and CST IV did not show a significant variation of the immune basal pattern.

### 3.4. Infection Follow-Up

Samples from women who tested positive for oncogenic viruses were recovered after eight months from laboratory diagnosis for additional gynecological and microbiological evaluations, as indicated by clinical protocol. Clinical examinations showed no sign of cervical lesions.

Microbiological analysis showed no residual viral DNA in 15/29 (51.7%) of the samples. To note, clearance of the virus was more frequently observed in CST I group dominated by *L. crispatus* (5/7 samples). Conversely, persistent infections due to HPV and *Polyomavirus*-HPV co-infections were confirmed in CST III (9/12) and CST IV (5/6) groups, which are considered to indicate a transitional state into a dysbiotic and a dysbiotic-BV like composition, respectively.

## 4. Discussion

One of the most interesting aspects of the virome world is its interaction with resident bacteria, probably influencing viral infectivity and thus the infection outcome. These events, playing a central role in shaping innate and adaptive immune defenses, are exerted during the early phase of viral entry, clinically corresponding to the asymptomatic phase of infection [19,20].

Despite increased attention has been focused on the complexity of vaginal microbiome and its role in women’s health, the modality of interaction between oncogenic viruses and local microenvironment is still incomplete.

A crucial point is the definition of the vaginal milieu that may influence the susceptibility and clearance of oncogenic viruses in women with an immune competent system, even if recent data suggest the vaginal dysbiosis as a risk factor for viral infections [8].

In order to optimize the complexity of the vaginal microbial profiles, the vaginal CSTs classification based on the central role of *Lactobacillus* spp. in orchestrating the vaginal composition and immune defense has been exploited in our study [1]. This study-approach has proved a valuable tool for a better characterization of the vaginal microbiome, as well as for providing direction on the events governing microbiome-virome and host immune response interactions.

We reinforced the concept that the species of *Lactobacillus* characterizing CSTs represent a key factor in influencing the modality of vaginal defense against viral infection through their direct involvement or by the modulation of the local immune response [12]. In the CST I, dominated by *L. crispatus*, the increase of the relative abundance of this *Lactobacillus*, as the main controlling infection defense mechanism, did not modify the basal immune profile. The antiviral activity of *L. crispatus* has been very recently reported in mammalian cell lines to be preventing HSV entry to cells [21] and blocking progression of HPV cervical lesions [22]. To corroborate these data, we demonstrated that 70% of women with a CST I profile cleared HPV and MCPyV infections in follow-up samples. Thus, suggesting a high potential activity of *L. crispatus* as a first-line defense against HPV and MCPyV which share similar pathogenic mechanisms, even if MCPyV only very recently has been proposed as a sexually transmitted microorganism [23].

In contrast, in CST II dominated by *L. gasseri* and in CST III by *L. iners*, a different type of host response resulting in the modulation of immune profile leading to the simultaneous increase of pro-inflammatory cytokines and the decrease of anti-inflammatory and anti-tumoral factors has been observed. In the CST IV, depleted of *Lactobacilli*, the increase of *Sneathia sanguinegens* and *Prevotella timonensis* was observed in infected samples. Our in vivo study suggests the low antiviral efficacy of *L. gasseri* and *L. iners*, in contrast with recent in vitro data pointing to a protective anti-tumoral effect exerted by all *Lactobacilli* species colonizing the cervical epithelium [24].

Previous studies have indicated the vaginal dysbiosis as a risk factor for HPV infection, although they cannot exclude reverse causation of HPV in altering the vaginal microbiome. A higher level of *Sneathia* spp. has been observed in HPV-positive women and associated with all stages of cervical carcinogenesis [25]. More recently, Mitra et al. identified *S. sanguinegens* to be more prevalent in women with high-grade dysplasia compared to women with a low-grade dysplasia [26] while other authors found an association of *Prevotella* spp. with symptomatic HPV infection [27].

We speculate that the composition of the vaginal microbiome “per se” is not a risk factor for the acquisition of viral infection but it seems to influence the viral clearance.

In our cohort, we detected an increase of the relative abundance of *Prevotella timonensis* and *Sneathia sanguinegens* in women infected with HPV or *Polyomavirus* during the asymptomatic phase of infection, event probably facilitated by a steady state level of pro-inflammatory status characterizing the vaginal environment of these women. Thus, it seems that both HPV and *Polyomaviruses* may promote their active colonization of the mucosal microenvironment through immune evasion strategies, mainly directed towards down-regulating the pro-inflammatory response. Indeed, a decreased concentration of the pro-inflammatory IL-15, IL-7, and the anti-inflammatory IL-9 was observed. The down-regulation of these molecules, which are first-line immune components for fighting viruses, prevents basically the growth and the activity of monocytes and T cells and, most importantly, inhibits their immunostimulatory and immunomodulatory effects [28,29,30]. Moreover, there is wide evidence supporting that host immunological features and local HPV-induced immunosuppressive environment constitute crucial points in establishing a persistent infection [31]. This feature fits our in vivo data in showing a low rate of viral clearance in women with a CST III or CST IV vaginal microbiome.

Newly, the detection of MCPyV in the vaginal mucosa, as unique strain or in samples co-infected with HPV, reinforces a role for this oncogenic *Polyomavirus* as a sexually transmitted pathogen, although no significant association between MCPyV and cervical cancer has been already demonstrated. To note, HPV and MCPyV co-occurrence was detected mainly in samples with molecular signatures of mucosal inflammation, thereby indicating a similarity in the pathogenic pathway.

As already disclosed by several studies, the total *Lactobacilli* load in the vagina is not an accurate parameter for asserting health status. Based on this observation, the introduction of CSTs has led to the comprehension that species of *Lactobacilli* differently exert a protective role in oncogenic viral infections. In addition, we highlighted that the microbiome heterogeneity is not predictive for HPV and *Polyomavirus* susceptibility, which, on the contrary, seems to benefit from the steady-state immune-bacterial crosstalk for establishing persistent infections. A main limitation of our study must be identified in the small study cohort. For this reason, we believe that further studies are needed to give more insights into the cross-talk between vaginal bacteriome and oncogenic virome.

## Figures and Tables

**Figure 1 microorganisms-07-00414-f001:**
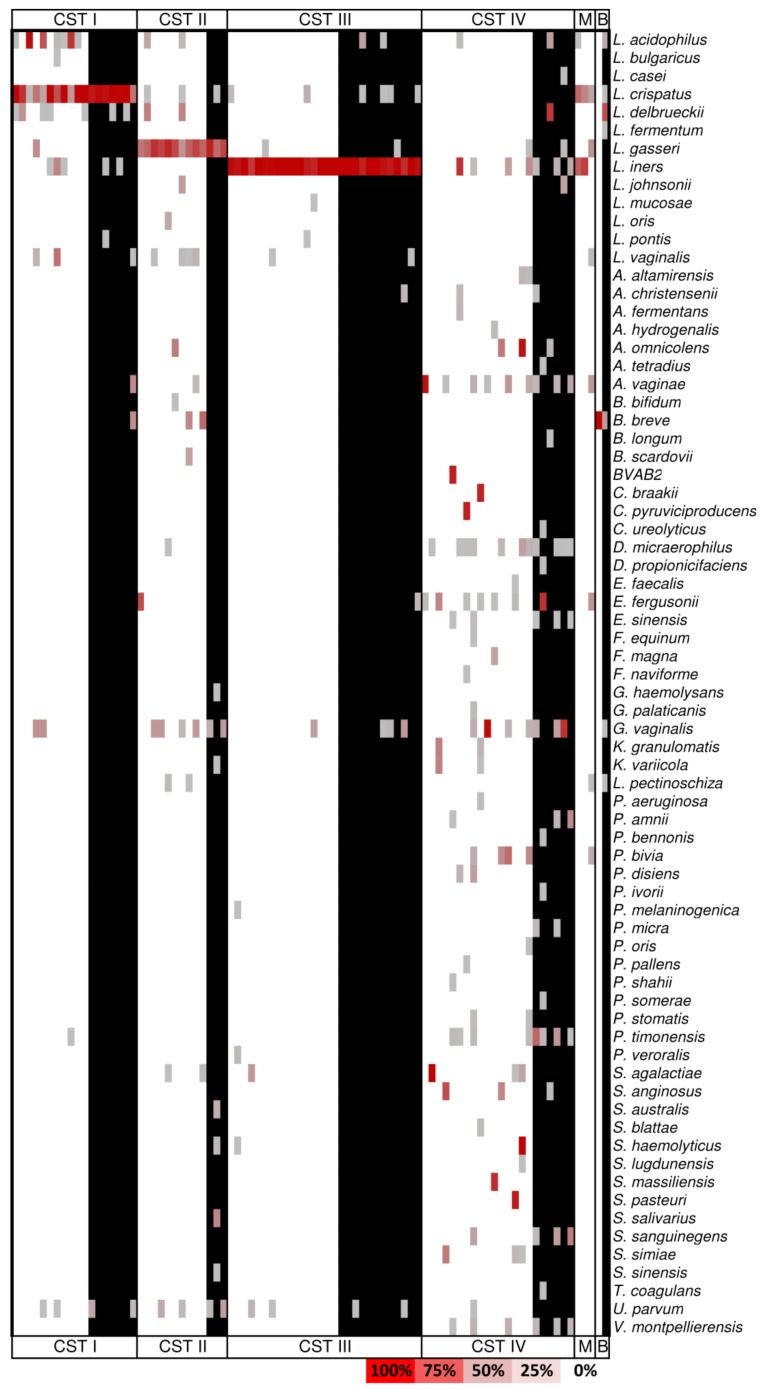
The microbiome composition according to the community state types (CSTs) classification. The heat map shows the relative abundances of the bacteria identified in the CSTs groups: CST I is dominated by *L. crispatus*, CST II by *L. gasseri*, CST III by *L. iners*, and CST IV is depleted of *Lactobacilli*. The relative abundances were calculated on the rarefied_otu_table.biom (5000 reads/sample) by means of the summarize_taxa.py script (QIIME 1.9.1). Within each CST, the black bars identify the viral DNA positive samples. Abbreviations: M: mixed CST; B: *Bifidobacteria*.

**Figure 2 microorganisms-07-00414-f002:**
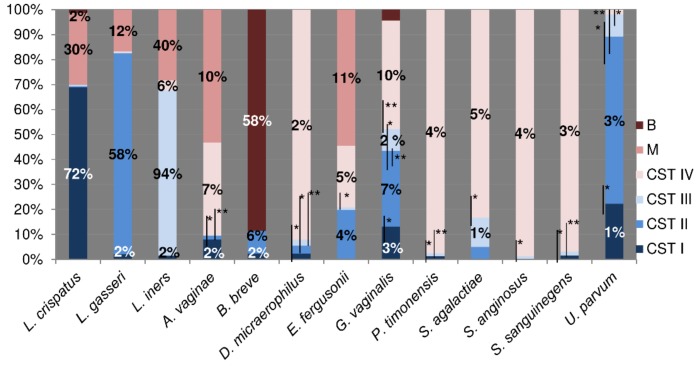
The bacterial differences among CSTs. The figure shows the significantly modulated bacteria among CSTs (*p* value < 0.05). Differences were calculated by means of a non-parametric T-test for pairwise comparisons among CSTs (Quantitative Insights Into Microbial Ecology, QIIME, 1.9.1). White bars show significant comparisons. * *p* < 0.05; ** *p* < 0.01.

**Figure 3 microorganisms-07-00414-f003:**
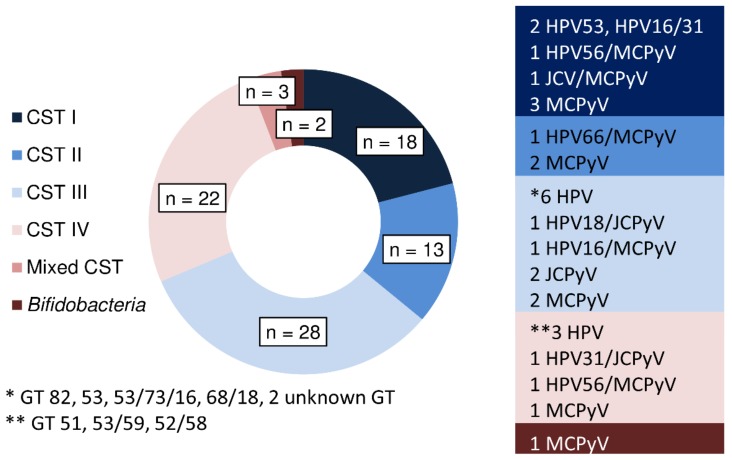
The CSTs distribution. The grouping of samples according to the vaginal community state types (CSTs) and the distribution of viruses within each CST. Abbreviations: GT: genotype; n = number of samples.

**Figure 4 microorganisms-07-00414-f004:**
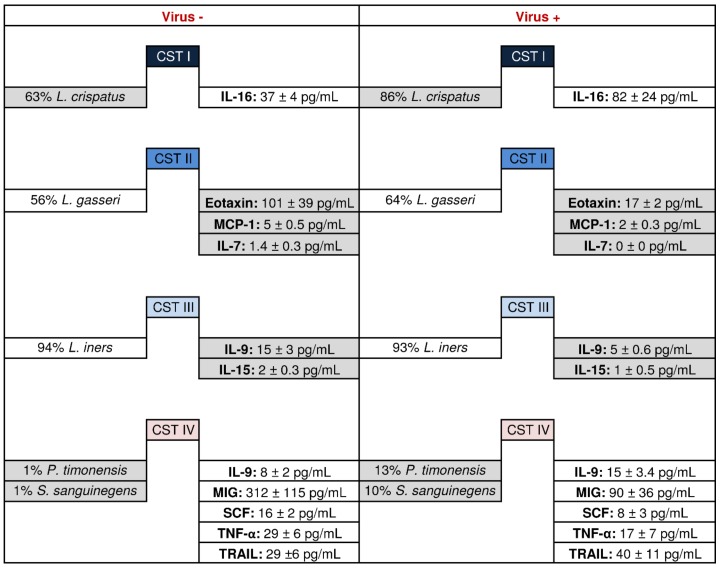
The impact of viruses on the CSTs. The figure shows the significantly modulated aspects between viral DNA negative vs. positive samples within each CST (*p* value < 0.05). Mixed CST and *Bifidobacteria* groups are missing as the first group did not show viral DNA while the latter had only one sample positive for viral DNA. Microbial differences were calculated by means of a non-parametric T-test for pairwise comparisons among CSTs using QIIME 1.9.1 while immunological differences were calculated using GraphPad Prism v. 5. The grey cells highlight the significant comparisons.

**Table 1 microorganisms-07-00414-t001:** Comparison of bacterial diversity among CSTs. Bacterial alpha diversity values are given as the mean ± standard deviation at a rarefaction depth of 5000 sequences per sample. Alpha diversity was compared among CSTs and, within each CST, between viral DNA positive and negative samples by means of a non-parametric t-test using the compare_alpha_diversity.py script of QIIME. a= *p* < 0.05 for CST III vs CST I and for CST III vs CST IV.

	CST I (18)		CST II (13)		CST III (28)		CST IV (22)		Mixed CST (3)		*Bifidobacteria* (2)	
**Chao1**	300 ± 106		282 ± 123		181 ± 103		274 ± 108		252 ± 147		281 ± 131	
Observed otus	114 ± 40		110 ± 43		70 ± 37^a^		125 ± 40		104 ± 54		99 ± 125	
PD whole tree	10 ± 3		9 ± 3		7 ± 3		10 ± 3		9 ± 3		10 ± 2	
	Virus		Virus		Virus		Virus		Virus		Virus	
	+ (7)	- (11)	+ (3)	- (10)	+ (12)	- (16)	+ (6)	- (16)	+	-	+ (1)	- (1)
Chao1	294 ± 123	305 ± 94	177 ± 90	314 ± 113	190 ± 85	174 ± 116	252 ± 133	282 ± 95	Na		413	150
Observed otus	114 ± 45	114 ± 36	82 ± 37	66 ± 42	73 ± 29	66 ± 42	124 ± 59	125 ± 31			129	69
PD whole tree	10 ± 3	10 ± 3	7 ± 2	10 ± 3	8 ± 2	7 ± 3	9 ± 3	9 ± 3			12	8

**Table 2 microorganisms-07-00414-t002:** The impact of viruses on the immunological differences among CSTs. The figure shows the significantly modulated immune soluble factors between viral DNA negative *vs* positive samples within each CST (*p* value < 0.05). Mixed CST and *Bifidobacteria* groups are missing as the first group did not show viral DNA while the latter had only one sample positive for viral DNA. Differences were calculated by means of a non-parametric T test for pairwise comparisons among CSTs (GraphPad Prism v. 5). Value are given as mean (pg/mL) ± standard error of the mean (SEM). **p* < 0.05, ** *p* < 0.01, empty cell= *p* value not significant. Bold = higher value within each CST.

	CST I		CST II		CST III		CST IV	
Cytokines	Virus +	Virus -	*p*	Virus +	Virus -	*p*	Virus +	Virus -	*p*	Virus +	Virus -	*p*
IL-1α	28 ± 10	77 ± 41		26 ± 23	88 ± 56		54 ± 13	144 ± 45		289 ± 158	188 ± 80	
IL-1β	21 ± 12	18 ± 6		47 ± 46	11 ± 5		39 ± 22	121 ± 56		77 ± 45	309 ± 142	
IL-1ra	2 × 10^5^ ± 8 × 10^5^	2.3 × 10^5^ ± 7 × 10^5^		3.4 × 10^5^ ± 3 × 10^5^	3.6 × 10^5^ ± 6 × 10^4^		2 × 10^5^ ± 3 × 10^4^	4 × 10^5^ ± 9 × 10^4^		5 × 10^5^ ± 9.6 × 10^4^	3.2 × 10^5^ ± 4.5 × 10^4^	
IL-2	2 ± 1	2 ± 0.7		1.3 ± 0.9	4 ± 1		0.6 ± 0.3	2 ± 0.6		3 ± 1.6	3 ± 0.7	
IL-2ra	91 ± 33	28 ± 4		36 ± 23	49 ± 26		35 ± 9	46 ± 11		33 ± 9	40 ± 9	
IL-3	132 ± 21	93 ± 8		57 ± 9	87 ± 9		100 ± 16	99 ± 12		80 ± 12	93 ± 8	
IL-4	0.8 ± 0.3	0.7 ± 0.1		0.3 ± 0.3	0.8 ± 0.2		0.5 ± 0.1	0.8 ± 0.1		0.6 ± 0.5	0.5 ± 0.1	
IL-5	0 ± 0	0 ± 0		0 ± 0	0.1 ± 0.1		0 ± 0	0.1 ± 0.1		0.7 ± 0.5	0 ± 0	
IL-6	2.6 ± 0.3	3.4 ± 1		2.6 ± 1.7	4 ± 0.9		3 ± 0.7	5 ± 1		2 ± 1	3.7 ± 1	
IL-9	9 ± 1.5	8 ± 2		12 ± 7	10 ± 2		5 ± 0.6	15 ± 3	**	15 ± 3.4	8 ± 1.6	
IL-10	1.3 ± 0.2	1.3 ± 0.1		1 ± 0	2 ± 0.1		1.6 ± 0.3	1.4 ± 0.2		1.5 ± 0.3	1.9 ± 0.3	
IL-12p40	357 ± 38	332 ± 23		130 ± 26	305 ± 25		305 ± 39	301 ± 31		232 ± 63	303 ± 22	
IL-12p70	6 ± 2	5.7 ± 1		2.6 ± 1	6 ± 1.4		5.7 ± 1	14 ± 4		6 ± 2.5	6 ± 1	
IL-13	1 ± 0	1 ± 0		1 ± 0	1.1 ± 0.1		1 ± 0.1	1 ± 0.1		1 ± 0.3	1.1 ± 0.1	
IL-15	1.4 ± 0.6	1.3 ± 0.5		1.7 ± 0.7	2 ± 0.6		1 ± 0.5	2 ± 0.3	*	2.8 ± 1.1	1.6 ± 0.5	
IL-16	81.6 ± 24.2	36.5 ± 3.9		244 ± 194	51.3 ± 19.9		36.9 ± 5.8	52.7 ± 10.2		40.8 ± 8.82	39.7 ± 3.2	
IL-17	7 ± 2	8.3 ± 2.5		6 ± 3	9 ± 3		8 ± 2	9.4 ± 2.2		7.3 ± 1.6	6 ± 0.8	
IL-18	24 ± 2.8	710 ± 568		66 ± 32	340 ± 196		125 ± 84	198 ± 83		393 ± 238	1E3 ± 506	
IFN-α2	29 ± 6	28 ± 11		14 ± 3	20 ± 5		18 ± 3.8	20 ± 3		17 ± 3	18 ± 2	
IFN-γ	34 ± 11	22 ± 4		14 ± 6	29 ± 5		18 ± 3.5	31 ± 5		32 ± 16	24 ± 4	
LIF	19 ± 2	20 ± 3		10 ± 2	16 ± 1.6		19 ± 2	19 ± 2.3		21 ± 6	23 ± 2.4	
MIF	1.3 × 10^3^ ± 1 × 10^3^	822 ± 665		6 × 10^3^ ± 4 × 10^3^	1.3 × 10^3^ ± 902		1 × 10^3^ ± 685	1.7 × 10^3^ ± 1 × 10^3^		3.4 × 10^3^ ± 2 × 10^3^	3 × 10^3^± 888	
SCF	27 ± 8	15 ± 3		26 ± 24	11 ± 2		18 ± 5	18 ± 5		8 ± 3.2	16 ± 2.5	
TNF-α	13 ± 1.6	12 ± 1.3		8 ± 3	17 ± 2		13 ± 3.4	20 ± 3		17 ± 7	29 ± 6	
TNF-β	14 ± 4	10 ± 0.4		7.3 ± 0.3	9.4 ± 0.3		13 ± 3	11 ± 2		10 ± 0.8	10 ± 0.5	
TRAIL	26 ± 2.4	21 ± 4.3		37 ± 13	18 ± 2.1		19 ± 1.6	23 ± 3.3		40 ± 11	29 ± 6	
	CST I		CST II		CST III		CST IV	
Chemokines	Virus +	Virus -	*p*	Virus +	Virus -	*p*	Virus +	Virus -	*p*	Virus +	Virus -	*p*
cTACK	101 ± 24	66 ± 10		37 ± 12	79 ± 15		70 ± 21	58 ± 15		62 ± 26	80 ± 10	
Eotaxin	52 ± 9	83 ± 41		17 ± 2	101 ± 39	**	83 ± 37	170 ± 65		54 ± 29	55 ± 13	
GRO- α	182 ± 108	632 ± 280		347 ± 318	521 ± 244		911 ± 248	959 ± 275		190 ± 102	907 ± 226	
IL-8	144 ± 59	148 ± 33		294 ± 203	171 ± 43		207 ± 50	519 ± 188		745 ± 502	454 ± 99	
IP-10	9 × 10^3^ ± 7 × 10^3^	2.5 × 10^4^ ± 2 × 10^4^		1.4 × 10^3^ ± 1.2 × 10^3^	1.3 × 10^3^ ± 399		1 × 10^4^ ± 5.7 × 10^3^	4.5 × 10^4^ ± 3 × 10^4^		324 ± 129	1.9 × 10^4^ ± 1.3 × 10^4^	
MCP-1	5 ± 0.5	10 ± 3.4		2.3 ± 0.3	5.5 ± 05	*	8 ± 2	6.3 ± 1.2		4 ± 1	5.3 ± 0.4	
MCP-3	9 ± 2	6 ± 0.4		5.7 ± 0.7	6.5 ± 1		6.3 ± 1	7 ± 1		7.2 ± 1.3	7.5 ± 0.7	
MIG	453 ± 257	288 ± 133		111 ± 65	149 ± 42		268 ± 170	724 ± 367		90 ± 36	312 ± 115	
MIP-1 α	1.1 ± 0.1	2 ± 0.5		1.3 ± 0.3	1.1 ± 0.1		1.5 ± 0.3	2.2 ± 0.6		2.2 ± 1	2.7 ± 0.5	
MIP-1 β	7 ± 2.5	21 ± 9		40 ± 38	14 ± 8		8 ± 3	40 ± 13		25 ± 11	32 ± 9	
RANTES	7.4 ± 2	8.3 ± 2.2		2.3 ± 0.3	26 ± 19		16 ± 9	7 ± 1.5		15 ± 6	28 ± 9	
SDF-1 α	147 ± 42	110 ± 22		77 ± 12	99 ± 19		130 ± 30	108 ± 14		101 ± 25	117 ± 12	
	CST I		CST II		CST III		CST IV	
Growth factors	Virus +	Virus -	*p*	Virus +	Virus -	*p*	Virus +	Virus -	*p*	Virus +	Virus -	*p*
FGF- β	23 ± 5	24 ± 4		10 ± 2	22 ± 4.5		21.6 ± 3.4	22 ± 5		15 ± 4.1	17 ± 1.4	
G-CSF	271 ± 172	212 ± 99		409 ± 384	89 ± 33		162 ± 57	686 ± 331		105 ± 79	116 ± 41	
GM-CSF	93 ± 9	110 ± 7.7		68 ± 21	96 ± 6.6		91 ± 9.5	100 ± 10		81 ± 10	77 ± 6	
HGF	172 ± 111	223 ± 149		387 ± 189	100 ± 35		235 ± 68	813 ± 402		300 ± 99	252 ± 95	
IL-7	1 ± 0.3	1.5 ± 0.7		0 ± 0	1.4 ± 0.3	*	0.6 ± 0.2	1.3 ± 0.5		0.8 ± 0.6	0.9 ± 0.3	
M-CSF	29 ± 8.3	29 ± 4.5		8.3 ± 3.3	27 ± 4		41 ± 5.2	44 ± 6.6		34 ± 13	33 ± 4	
PDGF- β	16 ± 3	35 ± 13		10 ± 4.6	30 ± 6.4		28 ± 6.2	27 ± 7.2		17 ± 4	19 ± 5	
SCGF- β	0 ± 0	0 ± 0		0 ± 0	0 ± 0		0 ± 0	0 ± 0		0 ± 0	0 ± 0	
VEGF	53 ± 24	47 ± 12		29 ± 16	47 ± 16		57 ± 12	131 ± 50		58 ± 26	81 ± 23	
β-NGF	3 ± 0.7	2 ± 0.3		2 ± 1	2 ± 0.5		1.8 ± 0.5	2.3 ± 0.5		2.6 ± 0.9	2.6 ± 0.4

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
