# Peer review of "Oncogenic Virome Benefits from the Different Vaginal Microbiome-Immune Axes"

_microorganisms, 2019, doi:10.3390/microorganisms7100414_

Round 1

Reviewer 1 Report

The authors evaluated the impact of oncoviruses on the vaginal community as well as the immune environment. This is an interesting study that could potentially improve our knowledge about the role of oncoviral infections in respect with vaginal bacteria and immune response. The manuscript is well written but will still benefit from minor language editing.

There are some points below that the authors should address.

Figure 1 image quality should be improved what are the potential limitations in this study? These should be included to the discussion section

Author Response

REVIEWER #1

Point 1. Figure 1 image quality should be improved

Response. We improved Figure 1 image quality.

Point 2. what are the potential limitations in this study? These should be included to the discussion section

                Response. We stated the potential limitations of this study in the discussion section (please, see lines 314-316)

Reviewer 2 Report

Giuseppina and colleagues in manuscript titled Oncogenic virome benefits from the different vaginal microbiome-immune axis” set out to explore the effect of vaginal microbiome and immune system to understand the role of oncogenic viruses in vaginal cancer such as cervical cancer.

Minor:

Citations: For example-Citation 2 is not appropriate. Use proper citations. I suggest revisiting all the citations to ensure that the citations are used appropriately. Important relevant papers are missed.

For example citation 2/3 Rx: https://www.ncbi.nlm.nih.gov/pmc/articles/PMC5164950/

Line 42 may need rephrasing so as to point what specific virus authors want to mention because small DNA element only refers to probably HPV and latter author cite Pandya et al for citation 11. Also, follow-up sentence also need rephrasing whether the infections should be latent, pseudo-latent or lytic?

Lines 50-52: Although persistent infection is important determinant of tumorigenesis, it is not the only determinant. Whether it is latent or lytic? Also, for example in case of HPV see this: https://www.ncbi.nlm.nih.gov/pmc/articles/PMC5164950/

The criteria on basis of which the subjects (aka demographics of study cohorts)were chosen should be tabulated. The table should indicate the umber of subject in each category that was initially used to classify the cohort.

Major:

How many of these patients were actually positive for any types of cancers? What are the differences in their microbial load (microbiome or virome) as compared to healthy individual?

Are there any observations made in study that will point out to a positive or negative association between presence of HPV and a “trigger” of carcinogenesis

+

Author Response

REVIEWER #2

Point 1. Citations: For example-Citation 2 is not appropriate. Use proper citations. I suggest revisiting all the citations to ensure that the citations are used appropriately. Important relevant papers are missed. For example citation 2/3 Rx: https://www.ncbi.nlm.nih.gov/pmc/articles/PMC5164950/ 

                Response. Thanks for your comment. We have changed and revisited all the citations.

Point 2. Line 42 may need rephrasing so as to point what specific virus authors want to mention because small DNA element only refers to probably HPV and latter author cite Pandya et al for citation 11. Also, follow-up sentence also need rephrasing whether the infections should be latent, pseudo-latent or lytic?

                Response. Thanks for your comment. We have changed the text accordingly (please see lines 42-47)

Point 3. Lines 50-52: Although persistent infection is important determinant of tumorigenesis, it is not the only determinant. Whether it is latent or lytic? Also, for example in case of HPV see this: https://www.ncbi.nlm.nih.gov/pmc/articles/PMC5164950/

                Response. We accept your comment and we have changed the text accordingly (please see lines 50-53).

Point 4. The criteria on basis of which the subjects (aka demographics of study cohorts)were chosen should be tabulated. The table should indicate the number of subject in each category that was initially used to classify the cohort.

                Response. We believe that a table would not add much information to the inclusion criteria described in the materials and methods section (please see lines 76-82).

Major:

Point 5. How many of these patients were actually positive for any types of cancers? What are the differences in their microbial load (microbiome or virome) as compared to healthy individual?

                Response. All the selected women included in the study were free from any cervical lesions and history of cancers, thus considered as healthy. The aim of this observational study was to evaluate the presence of silent oncoviral infections to individuate the group of women at high risk of infection.

The techniques used for microbiome (Ion Torrent Platform) and virome (Luminex Platform) profiling are not quantitative. Nevertheless, we observed significant changes in vaginal microbiome profiles in infected women when compared to negative ones (as shown in Fig 4).

Point 6. Are there any observations made in study that will point out to a positive or negative association between presence of HPV and a “trigger” of carcinogenesis

                Response. Although in our cohort HPV infected women did not develop intraepithelial lesions during the follow-up of the study, the persistence of viral infection reinforced the hypothesis of the role of the microbiome composition in influencing the host immune response. Specifically, in infected samples, we observed the presence of pathogenic bacteria, the down-regulation of cytokines with antiviral activity, and the up-regulation of pro-inflammatory molecules. This chronic low-level inflammation (the women are all asymptomatic) could represent an important trigger of viral persistence and tissue damage.